# Character-Level Language Modeling with Recurrent Highway Hypernetworks

**Joseph Suarez**
Stanford University
joseph15@stanford.edu

## Abstract

We present extensive experimental and theoretical support for the efficacy of recurrent highway networks (RHNs) and recurrent hypernetworks complimentary to the original works. Where the original RHN work primarily provides theoretical treatment of the subject, we demonstrate experimentally that RHNs benefit from far better gradient flow than LSTMs in addition to their improved task accuracy. The original hypernetworks work presents detailed experimental results but leaves several theoretical issues unresolved–we consider these in depth and frame several feasible solutions that we believe will yield further gains in the future. We demonstrate that these approaches are complementary: by combining RHNs and hypernetworks, we make a significant improvement over current state-of-the-art character-level language modeling performance on Penn Treebank while relying on much simpler regularization. Finally, we argue for RHNs as a drop-in replacement for LSTMs (analogous to LSTMs for vanilla RNNs) and for hypernetworks as a de-facto augmentation (analogous to attention) for recurrent architectures.

## 1 Introduction and related works

Recurrent architectures have seen much improvement since their inception in the 1990s, but they still suffer significantly from the problem of vanishing gradients [1]. Though many consider LSTMs [2] the de-facto solution to vanishing gradients, in practice, the problem is far from solved (see Discussion). Several LSTM variants have been developed, most notably GRUs [3], which are simpler than LSTM cells but benefit from only marginally better gradient flow. Greff et al. and Britz et al. conducted exhaustive (for all practical purposes) architecture searches over simple LSTM variants and demonstrated that none achieved significant improvement [4] [5]–in particular, the latter work discovered that LSTMs consistently outperform comparable GRUs on machine translation, and no proposed cell architecture to date has been proven significantly better than the LSTM. This result necessitated novel approaches to the problem.

One approach is to upscale by simply stacking recurrent cells and increasing the number of hidden units. While there is certainly some optimal trade off between depth and cell size, with enough data, simply upscaling both has yielded remarkable results in neural machine translation (NMT) [6].[1] However, massive upscaling is impractical in all but the least hardware constrained settings and fails to remedy fundamental architecture issues, such as poor gradient flow inherent in recurrent cells [8]. We later demonstrate that gradient issues persist in LSTMs (see Results) and that the grid-like architecture of stacked LSTMs is suboptimal.

The problem of gradient flow can be somewhat mitigated by the adaptation of Batch Normalization [9] to the recurrent case [10] [11]. While effective, it does not solve the problem entirely and also imposes significant overhead in memory and thus in performance, given the efficiency of parallelization over minibatches. This is often offset by a reduction in total epochs over the data required, but recurrent architectures with better gradient flow could ideally provide comparable or better convergence without reliance upon explicit normalization.

Zilly et al. recently proposed recurrent highway networks (RHNs) and offered copious theoretical support for the architecture's improved gradient flow [12]. However, while the authors provided mathematical rigor, we believe that experimental confirmation of the authors' claims could further demonstrate the model's simplicity and widespread applicability. Furthermore, we find that the discussion of gradient flow is more nuanced than presented in the original work (see Discussion).

Ha et al. recently questioned the weight-sharing paradigm common among recurrent architectures, proposing hypernetworks as a mechanism for allowing weight drift between timesteps [13]. This consideration is highly desirable, given the successes of recent convolutional architectures on language modeling tasks [14] [15], which were previously dominated by recurrent architectures.

Both RHNs and hypernetworks achieved state-of-the-art (SOTA) on multiple natural language processing (NLP) tasks at the time. As these approaches address unrelated architectural issues, it should not be surprising that combining them yields SOTA on Penn Treebank [16] (PTB), improving significantly over either model evaluated individually. We consider both RHNs and hypernetworks to be largely overlooked in recent literature on account of apparent rather than extant complexity. Furthermore, the original RHN work lacks sufficient experimental demonstration of improved gradient flow; the original hypernetworks work lacks theoretical generalization of their weight-drift scheme. We present experimental results for RHNs complementary to the original work's theoretical results and theoretical results for hypernetworks complementary to the original work's experimental results.

Founded on these results, our most important contribution is a strong argument for the utility of RHNs and hypernetworks, both individually and jointly, in constructing improved recurrent architectures.

## 2 Model architecture

### 2.1 Recurrent highway networks

We make a few notational simplifications to the original RHN equations that will later facilitate extensibility. We find it clearest and most succinct to be programmatic in our notation [2]. First, consider the GRU:

$$
\begin{aligned}
[h, t] &= x_i U + s_{i-1} W \qquad r = \tanh(x_i \hat{U} + (s_{i-1} \circ h)\hat{W}) \\
h, t &= \sigma(h), \sigma(t) \qquad s_i = (1 - t) \circ r + t \circ s_{i-1}
\end{aligned}
\tag{1}
$$

where $x \in \mathbb{R}^d$, $h, t, r, s_t \in \mathbb{R}^n$, and $U, \hat{U} \in \mathbb{R}^{d \times 2n}$, $W, \hat{W} \in \mathbb{R}^{n \times 2n}$ are weight matrices where $d, n$ are the input and hidden dimensions. $\sigma$ is the sigmoid nonlinearity, and $\circ$ is the Hadamard (elementwise) product. A one layer RHN cell is a simplified GRU variant:

$$
\begin{aligned}
[h, t] &= x_i U + s_{i-1} W \qquad s_i = (1 - t) \circ s_{i-1} + t \circ h \\
h, t &= \tanh(h), \sigma(t)
\end{aligned}
\tag{2}
$$

where the definitions from above hold. The RHN is extended to arbitrary depth by simply stacking this cell with new hidden weight matrices, with the caveat that $x_i U$ is omitted except at the input layer:

$$
\begin{aligned}
&RHNCell(x_i, s_{i-1}, l): \\
&[h, t] = \mathbb{1}\,[l = 0]\, x_i U + s_{i-1} W \qquad c, t = 1 - t, dropout(t) \\
&h, t = \tanh(h), \sigma(t) \qquad\qquad\quad s_i = c \circ s_{i-1} + t \circ h
\end{aligned}
\tag{3}
$$

where $l$ is the layer index, which is used as an indicator. We can introduce recurrent dropout [17] on $t$ across all layers with a single hyperparameter. We later demonstrate strong results without the need for more complex regularization or layer normalization. Finally, unlike stacked LSTMs, RHNs are structurally *linear*. That is, a depth $L$ RHN applied to a sequence of length $M$ can be unrolled to a simple depth $ML$ network. We restate this fact from the original work only because it is important to our analysis, which we defer to Results and Discussion.

## 2.2 Hypernetworks

We slightly alter the original notation of recurrent hypernetworks for ease of combination with RHNs. We define a *hypervector* $z$ as a linear upscaling projection applied to the outputs of a small recurrent network:

$$z(a) = W_p a \qquad (4)$$

where $a \in \mathbb{R}^h$ is the activation vector output by an arbitrary recurrent architecture, $W_p \in \mathbb{R}^{n \times h}$ is an upscaling projection from dimension $h$ to $n$, and $h \ll n$. The hypervector is then used to scale the weights of the main recurrent network by:

$$\widetilde{W}(z(a)) = z(a) \circ W \qquad (5)$$

where we overload $\circ$ as the element-wise product across columns. That is, each element of $z$ scales one column (or row, depending on notation) of $W$. As this constitutes a direct modification of the weights, hypernetworks have the interpretation of relaxing the weight sharing constraint implicit in RNNs.

## 2.3 Recurrent highway hypernetworks

We adapt hypernetworks to RHNs by directly modifying the RHN cell using (5):

$$
\begin{aligned}
RHNCellHyper&(x_i, s_{i-1}, l, z): \\
[h, t] &= \mathbb{1}\,[l=0]\, x_i \widetilde{U}(z) + s_{i-1}\widetilde{W}(z) \quad c, t = 1 - t, dropout(t) \\
h, t &= \tanh(h), \sigma(t) \qquad\qquad\qquad\quad s_i = c \circ s_{i-1} + t \circ h
\end{aligned}
\qquad (6)
$$

If $RHNCell$ and $RHNCellHyper$ had the same state sizes, we could simply stack them. However, as the hypernetwork is much smaller than the main network by design, we instead must upscale between the networks. Our final architecture at each timestep for layer $l$ can thus be written:

$$s_h = RHNCell(s_h, l) \quad z = [M_{pl}s_h, M_{pl}s_h] \quad s_n = RHNCellHyper(s_n, l, z) \qquad (7)$$

where $M_{pl} \in \mathbb{R}^{h \times n}$ is the upscale projection matrix for layer $l$ and $z$ is the concatenation of $M_{pl}s_h$ with itself. Notice the simplicity of this extension–it is at least as straightforward to extend RHNs as GRUs and LSTMs. Again, we use only simple recurrent dropout for regularization.

A few notes, for clarity and ease of reproduction: as the internal weight matrices of the main network have different dimensionality ($U_l \in \mathbb{R}^{d \times 2n}, W_l \in \mathbb{R}^{n \times 2n}$), we require the concatenation operation to form $z$ in (7). We find this works much better than simply using larger projection matrices. Also, $s_h, s_n$ in (7) are the hypernetwork and main network states, respectively. This may seem backwards from the notation above, but note that the hypernetwork is a standard, unmodified $RHNCell$; its outputs are then used in the main network, which is the modified $RHNCellHyper$.

# 3 Results (experimental)

## 3.1 Penn Treebank

Penn Treebank (PTB) contains approximately 5.10M/0.40M/0.45M characters in the train/val/test sets respectively and has a small vocabulary of 50 characters. There has recently been some controversy surrounding results on PTB: Jozefowicz et al. went as far to say that performance on such small datasets is dominated by regularization [18]. Radford et al. chose to evaluate language modeling performance only upon the (38GB) Amazon Product Review dataset for this reason [19].

Performance on large, realistic datasets is inarguably a better metric of architecture quality than performance on smaller datasets such as PTB. However, such metrics make comparison among models nearly impossible: performance on large datasets is non-standard because evaluation at this scale is infeasible in many research settings simply because of limited hardware access. While most models can be trained on 1-4 GPUs within a few weeks, this statement is misleading, as significantly more hardware is required for efficient development and hyperparameter search. We therefore emphasize the importance of small datasets for standardized comparison among models. Hutter is a medium sized task (approximately 20 times larger than PTB) that should be feasible in most settings (e.g. the original RHN and Hypernetwork works). We are only reasonably able to

Table 1: Comparison of bits per character (BPC) test errors on PTB. We achieve SOTA without layer normalization, improving over vanilla hypernetworks, which require layer normalization

| Model | Test | Val | Params (M) |
|---|---|---|---|
| LSTM | 1.31 | 1.35 | 4.3 |
| 2-Layer LSTM | 1.28 | 1.31 | 12.2 |
| 2-Layer LSTM (1125 hidden, ours) | – | 1.29 | 15.6 |
| HyperLSTM | 1.26 | 1.30 | 4.9 |
| Layer Norm LSTM | 1.27 | 1.30 | 4.3 |
| Layer Norm HyperLSTM | 1.25 | 1.28 | 4.9 |
| Layer Norm HyperLSTM (large embed) | 1.23 | 1.26 | 5.1 |
| 2-Layer Norm HyperLSTM, 1000 units | 1.22 | 1.24 | 14.4 |
| Recurrent Highway Network (ours) | - | 1.24 | 14.0 |
| HyperRHN (ours) | 1.19 | 1.21 | 15.5 |

evaluate on PTB due to a strict hardware limitation of two personally owned GPUs. We therefore take additional precautions to ensure fair comparison:

First, we address the critiques of Jozefoqicz et al. by avoiding complex regularization. We use only simple recurrent dropout with a uniform probability across layers. Second, we minimally tune hyperparameters as discussed below. Finally, we are careful with the validation data and run the test set only once on our best model. We believe these precautions prevent overfitting the domain and corroborate the integrity of our result. Furthermore, SOTA performance with suboptimal hyperparameters demonstrates the robustness of our model.

## 3.2 Architecture and training details

In addition to our HyperRHN, we consider our implementations of a 2-Layer LSTM and a plain RHN below. All models, including hypernetworks and their strong baselines, are compared in Table 1. Other known published results are included in the original hypernetworks work, but have test bpc $\geq$ 1.27. We train all models using Adam [20] with the default learning rate 0.001 and sequence length 100, batch size 256 (the largest that fits in memory for our main model) on a single GTX 1080 Ti until overfitting becomes obvious. We evaluate test performance only once and only on our main model, using the validation set for early stopping.

Our data batcher loads the dataset into main memory as a single contiguous block and reshapes it to column size 100. We do not zero pad for efficiency and no distinction is made between sentences for simplicity. Data is embedded into a 27 dimensional vector. *We do not cross validate any hyperparameters except for dropout*.

We first consider our implementation of a 2-Layer LSTM with hidden dimension 1125, which yields approximately as many learnable parameters as our main model. We train for 350 epochs with recurrent dropout probability 0.9. As expected, our model performs slightly better than the slightly smaller baseline in the original hypernetworks work. We use this model in gradient flow comparisons (see Discussion)

As the original RHN work presents only word-level results for PTB, we trained a RHN baseline by simply disabling the Hypernetwork augmentation. Convergence was achieved in 618 epochs.

Our model consists of a recurrent highway hypernetwork with 7 layers per cell. The main network has 1000 neurons per layer and the hypernetwork has 128 neurons per layer, for a total of approximately 15.2M parameters. Both subnetworks use a recurrent dropout keep probability of 0.65 and no other regularizer/normalizer. We attribute our model's ability to perform without layer normalization to the improved gradient flow of RHNs (see Discussion).

The model converges in 660 epochs, obtaining test perplexity 2.29 (where cross entropy corresponds to $\log_e$ of perplexity) and 1.19 bits per character (BPC, $\log_2$ of perplexity), 74.6 percent accuracy. By epoch count, our model is comparable to a plain RHN but performs better. Training takes 2-3 days (fairly long for PTB) compared to 1-2 days for a plain RHN and a few hours for an LSTM. However, this comparison is unfair: all models require a similar number of floating point operations and differ

primarily in backend implementation optimization. We consider possible modifications to our model that take advantage of existing optimization in Results (theoretical), below.

Finally, we note that reporting of accuracy is nonstandard. Accuracy is a standard metric in vision; we encourage its adoption in language modeling, as BPC is effectively a change of base applied to standard cross entropy and is exponential in scale. This downplays the significance of gains where the error ceiling is likely small. Accuracy is more immediately comparable to maximum task performance, which we estimate to be well below 80 percent given the recent trend of diminishing returns coupled with genuine ambiguity in the task. Human performance is roughly 55 percent, as measured by our own performance on the task.

## 4 Results (theoretical)

Our final model is a direct adaptation of the original hypervector scaling factor to RHNs. However, we did attempt a generalization of hypernetworks and encountered extreme memory considerations that have important implications for future work. Notice that the original hypernetwork scaling factor is equivalent to element-wise multiplication by a rank-1 matrix (e.g. the outer product of $z$ with a ones vector, which does not include all rank-1 matrices). Ideally, we should be able to scale by any matrix at all. As mentioned by the authors, naively generating different scaling vectors for each column of the weight matrix is prohibitively expensive in both memory and computation time. We propose a low rank-$d$ update inspired by the thin singular value decomposition as follows:

$$\widetilde{W} = W \circ \sum_{i=1}^{d} u_i v_i^\top \tag{8}$$

Compared to the original scaling update, our variation has memory and performance cost linear in the rank of the update. As with the SVD, we would expect most of the information relevant to the weight drift scale to be contained in a relatively low-rank update. However, we were unable to verify this hypothesis due to a current framework limitation. All deep learning platforms currently assemble computation graphs, and this low rank approximation is added as a node in the graph. This requires memory equal to the dimensionality of the scaling matrix *per training example*!

The original hypernetworks update is only feasible because of a small mathematical trick: row-wise scaling of the weight matrix is equal to elementwise multiplication after the matrix-vector multiply. Note that this is a practical rather than theoretical limitation. As variations in the weights of the hypernetwork arise only as a function of variations in $u_i, v_i, W$, it is possible to define a custom gradient operation that does not need to store the low rank scaling matrices at each time step for backpropagation.

Lastly, we note that hypernetworks are a new and largely unexplored area of research. Even without the above addition, hypernetworks have yielded large improvements on a diverse array of tasks while introducing a minimal number of additional parameters. The only reason we cannot currently recommend hypernetworks as a drop-in network augmentation for most tasks (compare to e.g. attention) is another framework limitation. Despite requiring far fewer floating point operations than the larger main network, adding a hypernetwork still incurs nearly a factor of two in performance. This due to the extreme efficiency of parallelization over large matrix multiplies; the overhead is largely time spent copying data. We propose rolling the hypernetwork into the main network. This could be accomplished by simply increasing the hidden dimension by the desired hypernetwork dimension $h$. The first $h$ elements of the activation can then be treated as the hypervector. Note that this may require experimentation with matrix blocking and/or weight masking schemes in order to avoid linear interactions between the hypernetwork and main network during matrix multiplication.

The issues and solutions above are left as thought experiments; we prioritize our limited computational resources towards experimental efforts on recurrent highway networks. The theoretical results above are included to simultaneously raise and assuage concerns surrounding generalization and efficiency of hypernetworks. We see additional development of hypernetworks as crucial to the continued success of our recurrent model in the same manner that attention is a necessary, de-facto network augmentation in machine translation (and we further expect the gains to stack). Our model's strong language modeling result using a single graphics card was facilitated by the small size of PTB, which allowed us to afford the 2X computational cost of recurrent hypernetworks. We present methods

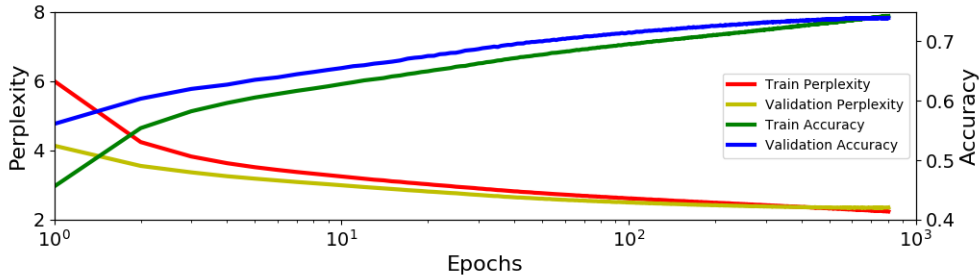

Figure 1: Visualization of hyper recurrent highway network training convergence

for optimizing the representational power and computational cost of hypernetworks; additional engineering will still be required in order to fully enable efficient training on large datasets.

## 5 Discussion (experimental)

### 5.1 Training time

We visualize training progress in Fig. 1. Notice that validation perplexity seems to remain below training perplexity for nearly 500 epochs. While the validation and test sets in PTB appear slightly easier than the training set, the cause of this artifact is that the validation loss is masked by a minimum 50-character context whereas the training loss is not (we further increase minimum context to 95 after training and observe a small performance gain), therefore the training loss suffers from the first few impossible predictions at the start of each example. The validation data is properly overlapped such that performance is being evaluated over the entire set.

It may also seem surprising that the model takes over 600 epochs to converge, and that training progress appears incredibly slow towards the end. We make three observations: first, we did not experiment with different optimizers, annealing the learning rate, or even the fixed learning rate itself. Second, as the maximum task accuracy is unknown, it is likely that gains small on an absolute scale are large on a relative scale. We base this conjecture on the diminishing gains of recent work on an absolute scale: we find that the difference between 1.31 (1 layer LSTM) and 1.19 bpc (our model) is approximately 71.1-74.6 percent accuracy. For reference, our improvement over the original hypernetworks work is approximately 1.0 percent (this figure is obtained from interpolation on the BPC scale). Third and finally, regardless of whether our second observation is true, our architecture exhibits similar convergence to a RHN and begins outperforming the 2-layer LSTM baseline before the latter converges.

### 5.2 Overview of visualizations

Our motivation in the visualizations that follow is to compare desirable and undesirable properties of our RHN-based model and standard recurrent models, namely stacked LSTMs. There are two natural gradient visualizations: within-cell gradients, which are averaged over time but not over all of the weight layers within the recurrent cell, and outside-cell gradients, which are averaged over internal weight layers but not over time. Time-averaged gradients are less useful to our discussion than the norms of raw weight layers; we therefore present these along with outside-cell gradient visualizations.

### 5.3 Cell visualizations

We visualize the 2-norms of the learned weight layers of our RHN-based model in Fig. 2 and of an LSTM baseline (2 layers, 1150 hidden units, recurrent dropout keep p=0.90, 15.6M parameters) in Fig. 3.

Notice that in the middle six layers (the first/last layers have different dimensionality and are incomparable) of the RHN block (Fig. 2), weight magnitude decreases with increasing layer depth. We view this as evidence for the iterative-refinement view of deep learning, as smaller updates are

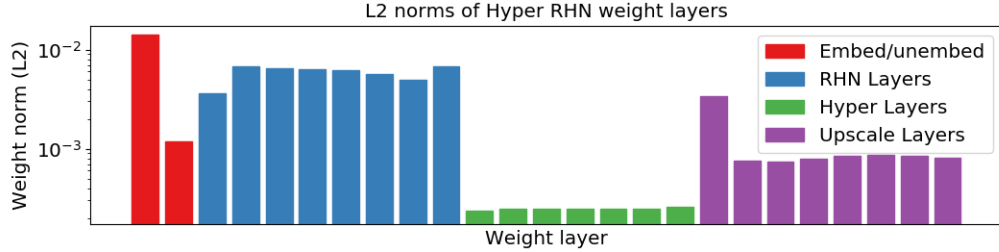

Figure 2: L2 norms of learned weights in our recurrent highway hypernetwork model. Increasing depth is shown from left to right in each block of layers. As dimensionality differs between blocks, the middle layers of each block are incomparable to the first/last layers, hence the disparity in norm.

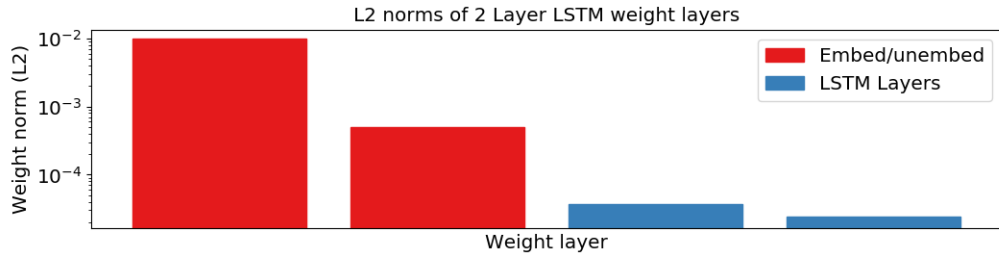

Figure 3: L2 norms of learned weights in our 2-layer LSTM baseline, with layer 1 left of layer 2.

applied in deeper layers. This is first evidence of this paradigm that we are aware of in the recurrent case, as similar statistics in stacked LSTMs are less conclusive because of horizontal grid connections. This also explains why performance gains diminish as RHN depth increases, as was noted in the original work.

### 5.4 Gradient visualizations over time

We consider the mean L2-norms of the gradients of the activations with respect to the loss at the final timestep. But first, an important digression: when should we visualize gradient flow: at initialization, during training, or after convergence? To our knowledge, this matter has not yet received direct treatment. Fig. 4 is computed at initialization and seems to suggest that RHNs are far inferior to LSTMs in the multilayer case, as the network cannot possibly learn in the presence of extreme vanishing gradients. This line of reasoning lacks the required nuance, which we discuss below.

## 6 Discussion (theoretical)

We address the seemingly inconsistent experimental results surrounding gradient flow in RHN.

First, we note that the LSTM/RHN comparison is unfair: multilayer LSTM/GRU cells are laid out in a grid. The length of the gradient path is equal to the sum of the sequence length and the number of layers (minus one); in an RHN, it is equal to the product. In the fair one layer case, we found that the RHN actually possesses far greater initial gradient flow. Second, these intuitions regarding vanishing gradients at initialization are incorrect. As shown in Fig. 5, gradient flow improves dramatically after training for just one epoch. By convergence, as shown in Fig. 6, results shift in the favor of RHNs, confirming experimentally the theoretical gradient flow benefits of RHNs over LSTMs.

Third, we address a potential objection. One might argue that while the gradient curves of our RHN based model and the LSTM baseline are similar in shape, the magnitude difference is misleading. For example, if LSTMs naturally have a million times smaller weights, then the factor of a hundred magnitude difference in Fig. 6 would actually demonstrate superiority of the LSTM. This is the reason for our consideration of weight norms in Fig. 2-3, which show that LSTMs have only 100 times smaller weights. Thus the gradient curves in Fig. 6 are effectively comparable in magnitude. However, RHNs maintain gradient flow equal to that of stacked LSTMs while having far greater

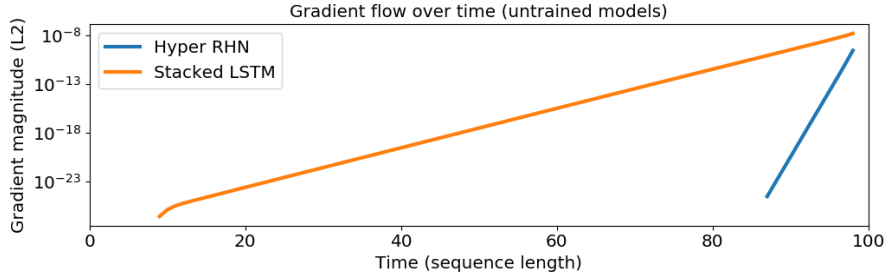

Figure 4: Layer-averaged gradient comparison between our model and an LSTM baseline. Gradients are computed at initialization at the input layer of each timestep with respect to the final timestep's loss. Weights are initialized orthogonally.

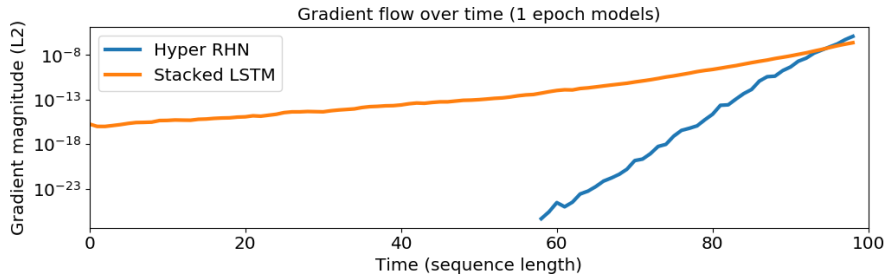

Figure 5: Identical to Fig. 4, but gradients are computed from models trained for one epoch.

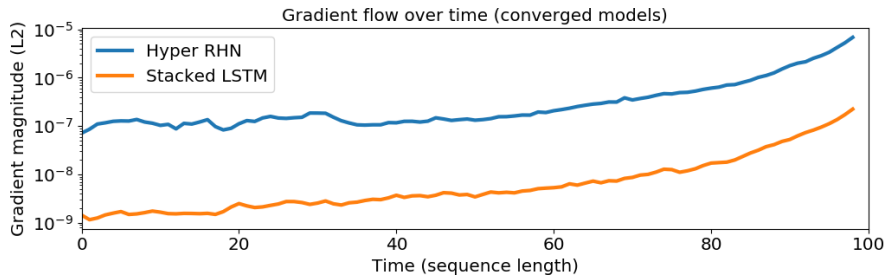

Figure 6: Identical to Fig. 4, but gradients are computed after convergence.

gradient path length, thus the initial comparison is unfair. We believe that this is the basis for the RHN's performance increase over the LSTM: RHNs allow much greater effective network depth without incurring additional gradient vanishing.

Fourth, we experimented with adding the corresponding horizontal grid connections to our RHN, obtaining significantly better gradient flow. With the same parameter budget as our HyperRHN model, this variant obtains 1.40 bpc–far inferior to our HyperRHN, though it could likely be optimized somewhat. It appears that long gradient paths are precisely the advantage in RHNs. We therefore suggest that gradient flow specifically along the deepest gradient path is an important consideration in architecture design: it provides an upper limit on effective network depth. It appears that greater effective depth is precisely the advantage in modeling potential of the RHN.

# 7 Conclusion

We present a cohesive set of contributions to recurrent architectures. First, we provide strong experimental evidence for RHNs as a simple drop-in replacement for stacked LSTMs and a detailed discussion of several engineering optimizations that could further performance. Second, we visualize

and discuss the problem of vanishing gradients in recurrent architectures, revealing that gradient flow significantly shifts during training, which can lead to misleading comparisons among models. This demonstrates that gradient flow should be evaluated at or near convergence; using this metric, we confirm that RHNs benefit from far greater effective depth than stacked LSTMs while maintaining equal gradient flow. Third, we suggest multiple expansions upon hypernetworks for future work that have the potential to significantly improve efficiency and generalize the weight-drift paradigm. This could lead to further improvement upon our architecture and, we hope, facilitate general adoption of hypernetworks as a network augmentation. Finally, we demonstrate effectiveness by presenting and open sourcing (code [3]) a combined architecture that obtains SOTA on PTB with minimal regularization and tuning which normally compromise results on small datasets.

## Acknowledgments

Special thanks to Ziang Xie, Jeremy Irvin, Dillon Laird, and Hao Sheng for helpful commentary and suggestion during the revision process.

## Footnotes

[1]For fair comparison, Google's NMT system does far more than upscaling and includes an explicit attentional mechanism [7]. We do not experiment with attention and/or residual schemes, but we expect the gains made by such techniques to stack with our work.

[2]Note that for purpose of clean alignment, equations are presented top to bottom, then left to right.

[3]github.com/jsuarez5341/Recurrent-Highway-Hypernetworks-NIPS

[14] Yann N Dauphin, Angela Fan, Michael Auli, and David Grangier. Language modeling with gated convolutional networks. *arXiv preprint arXiv:1612.08083*, 2016.

[15] Nal Kalchbrenner, Lasse Espeholt, Karen Simonyan, Aaron van den Oord, Alex Graves, and Koray Kavukcuoglu. Neural machine translation in linear time. *arXiv preprint arXiv:1610.10099*, 2016.

[16] Mitchell P Marcus, Mary Ann Marcinkiewicz, and Beatrice Santorini. Building a large annotated corpus of english: The penn treebank. *Computational linguistics*, 19(2):313–330, 1993.

[17] Stanislau Semeniuta, Aliaksei Severyn, and Erhardt Barth. Recurrent dropout without memory loss. *arXiv preprint arXiv:1603.05118*, 2016.

[18] Rafal Jozefowicz, Oriol Vinyals, Mike Schuster, Noam Shazeer, and Yonghui Wu. Exploring the limits of language modeling. *arXiv preprint arXiv:1602.02410*, 2016.

[19] Alec Radford, Rafal Jozefowicz, and Ilya Sutskever. Learning to generate reviews and discovering sentiment. *arXiv preprint arXiv:1704.01444*, 2017.

[20] Diederik Kingma and Jimmy Ba. Adam: A method for stochastic optimization. *arXiv preprint arXiv:1412.6980*, 2014.

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
