[Reviews · NeurIPS 2017]

Reviewer 1



Summary: The paper augments Recurrent Highway Networks (RHNs) with a "hypernetwork." Results are presented on character-level language modelling on the PTB benchmark where the authors show a small improvement over several HyperLSTM variants (SOTA as they say). Furthermore, they discuss the problem of vanishing gradients in their architecture and compare those gradients to those of a stacked LSTM. Pros: - good and reasonable results - reinforces the superiority of RHN over standard LSTM for language modelling on PTB using a gradient analysis. Cons: - In terms of new contributions, the paper seems not that strong. - The experiments are limited to character-level modelling only. The authors do mention hardware limitations but fail to provide e.g. word-level results for PTB. - While they compare with an HyperLSTM they do not compare ith a plain RHN. It's implicitly assumed that the hypernetwork augmentation will improve a comparable RHN architecture but this has not been shown. Perhaps one could quickly set up additional experiments to test this. - Layer Norm was not used in the architecture but since LN has been shown to lead to significant improvements in combination with Hypernetworks this could have been tested empirically. References: - One cannot cite only the vanishing gradient work of Bengio et al (1994) without citing the original, essentially identical work of Hochreiter (1991), simply because the 1994 paper does not cite the original work. One way around this is to cite only the 2001 paper, where Hochreiter is 1st author, and Bengio 2nd. Hochreiter, Sepp. Untersuchungen zu dynamischen neuronalen Netzen. Diploma thesis, TUM, pp. 91, 1991. - line 17-21: this is misleading - mention that vanilla LSTM consistently outperforms the cited LSTM variant called "GRU" on NLP and machine translation according to Google's cited analysis: https://arxiv.org/abs/1703.03906 Formulas: - The authors decided to use a notation different from the one of the authors of the Recurrent Highway Network and HyperNetworks paper and fail to properly define its meaning (e.g., h is not the state but s is). This makes it unnecessarily more difficult to understand their formulas. General recommendation: Slightly leaning towards accept, provided the comments are taken into account. Let's wait for the rebuttal phase.

Reviewer 2



The paper describes how to achieve state-of-the-art results on a small character prediction task. Some detailed comments below: - Title: First of all, I would not call work on PTB using characters "language modeling", it is character prediction. Yes, theoretically the same thing but actual language model applications typically use much bigger vocabularies and much more data, and this is where most of the problems in language modeling lie. Replacing "language modeling" by "character prediction" would be a more appropriate title - Results (experimental): I agree that PTB is OK to use as a database to test character prediction models. However, it would have been good to also test/show your findings on larger databases like for example the 1 Billion Google corpus or similar. Public implementations for training these relatively quickly on limited hardware exist and it would have made this a much stronger paper, as eventually I believe you hope that your findings will be in fact used for larger, real problems. To show in your paper that you can handle large problems is always better than finding reasons to not do it as you do in line 113-118 and in other places throughout the paper. - Results (experimental): You list accuracy/perplexity/bpc -- it would be good to define exactly how these are calculated to avoid confusion. - 3.2: 2-3 days of training for PTB is quite long, as you note yourself. It would have been nice to compare the different models you compare yourself over time by plotting perplexity vs. training time for all the models to compare how efficiently they can be trained. - 5.1: You say that you are almost certainly within a few percent of maximum performance on this task -- why? Maybe that is true but maybe it isn't. Diminishing gains on an absolute scale can have many reasons, including the one that possibly many people do not work on PTB for character prediction anymore. Also, the last sentence: Why is accuracy better than perplexity to show relative improvement? BTW, I agree that accuracy is a good measure in general (because everybody knows it is bounded between 0 and 100%), but perplexity is simply the standard measure for these kinds of predictions tasks. - 5.4: You say that how and when to visualize gradient flow has not been receiving direct treatment. Yes, true, but I would argue the main reason is that it is simply not so interesting -- what matters in training these networks is not a theoretical treatment of some definition of gradient flow but simply time to convergence for a given size model on a given task. This goes also back to my comment from above, what would have been more interesting is to show training time vs accuracy for several different networks and different sizes. - 6: Why can you compare gradient norms of completely different architectures? I would argue it is quite unclear whether gradients of different architectures have to have similar norms -- the error surface could look very different. Also, even if you can compare them, the learning algorithm matters a lot -- SGD vs Adam vs Rprop or whatnot have all a completely different effect given a certain gradient (in some cases only the sign matters etc.). So overall, I would argue that gradient flow is simply not important, it would be better to focus to convergence optimizing learning algorithms, hyperparameters etc.

Reviewer 3



This paper discusses a combination of recurrent highway networks (RHNs) and hypernetworks. Whereas simply joining 2 existing techniques may not seem pretty original, the paper does present some nice theoretical results for hypernetworks, nice experimental results for RHNs and a good amount of analysis. The authors refer to all the relevant papers and exhibit some interesting insights into models that have previously been explained in an overly complicated manner. I feel that this paper succeeds in what it intends to do: advocate RHNs as the default building block for recurrent architectures, rather than LSTMs. Some minor comments: * line 44: (e.g. on Penn Treebank [16] (PTB)) * line 58 and later occurrences: Lowercase "where" * equations 1-3: decide on s_t or s_i, don't mix * footnote 2: purposed -> purpose * equation 7: clarify M_{pl}, use math notation instead of duplicate function * equation 7: where do s_h and s_n come from? they are not introduced in equation 6... * figures 1, 2, 3: perhaps show later, as it is only discussed on the next page * figures 2 and 3: label the weight layers, as it is not clear what is the first and the last layer. figure 3 seems to suggest that the 4th bar corresponds to layer 1, yet for figure 2 you mention that the gradients increase during back propagation which seems to indicate the opposite * line 236: twice "the" * line 259: an -> a * line 285: LSTM